# Prevalence of depression and associated factors among adult cancer patients receiving chemotherapy during the era of COVID-19 in Ethiopia. Hospital-based cross-sectional study

Abebe Muche Belete[1]*, Anmut Alemagegn[2], Anemut Tilahun Mulu[3], Taklo Simeneh Yazie[4], Bekalu Bewket[5], Adisu Asefa[1], Wendimeneh Shibabaw Shiferaw[6]

1 Department of Biomedical Science, College of Health Science, Debre Berhan University, Debre Berhan, Ethiopia, 2 Oncology Clinic, Tikur Anbesa Specialized Hospital, Addis Ababa University, Addis Ababa, Ethiopia, 3 Department of Biomedical Science, College of Health Science, Debre Tabor University, Debre Tabor, Ethiopia, 4 Pharmacology and Toxicology Unit, Department of Pharmacy, College of Health Sciences, Debre Tabor University, Debre Tabor, Ethiopia, 5 Department of Nursing, College of Health Science, Injibara University, Injibara, Ethiopia, 6 Department of Nursing, College of Health Science, Debre Berhan University, Debre Berhan, Ethiopia

* abebemuche3@gmail.com

## Abstract

### Background

Depression is a major public health problem among cancer patients undergoing chemotherapy. It compromises patient outcomes, resulting in higher rates of mortality. Currently, there are little data on the prevalence of depression in Sub-Sharan countries. Therefore, the current study was done to assess the prevalence of depression among adult cancer patients and its associated factors in Ethiopia.

### Method

An institutional based cross-sectional study was conducted among 420 adult cancer patients from 1st March to April 30, 2021. Systematic random sampling technique was used to recruit participants. Depression was assessed using the Patient Health Questionnaire-9. The collected data were coded and entered into Epi-data version 4.2 and exported to SPSS version 25 for analysis. Bivariate binary logistic regression was performed to select factors that will be included in multivariate analysis, and variables with a p-value < 0.05 were included in multivariate analysis. In multivariate analysis, odds ratios and their 95% confidence intervals were computed and variables with p-value < 0.05 were considered to declare a significant association.

### Results

In this study, the prevalence of depression was 33.1% (95% CI = 0.2858, 0.3761). Minimal symptoms of depression, minor depression, moderate depression, moderate severe depression, and severe depression were found to be 272 (64.8%), 9 (2.1%), 104 (24.8%),

**Data Availability Statement:** All relevant data are within the paper and its Supporting Information files.

**Funding:** The authors received no specific funding for this work.

**Competing interests:** The authors have declared that no competing interests exist.

**Abbreviations:** AOR, Adjusted Odds Ratio; CI, Confidence Interval; COR, Crud Odds ratio; PHQ, Patient Health Questionnaire; SD, Standard Deviation; SPSS, Statistical Package for Social Sciences; TASH, Tikur Anbessa Specialized Hospital.

28 (6.7%), and 7 (1.7%), respectively. Those cancer patients who had unemployed status, sacked from jobs, and had stage IV cancer were most likely to develop depression, whereas patients who completed primary education, colon, prostate and cervical cancer were less likely to have depression.

## Conclusion

Depression is found to be a major public health concern for adult cancer patients in Ethiopia. To reduce the occurrence of depression among cancer patients, special attention is needed.

## Background

Worldwide, new cancer cases, and cancer deaths were estimated to be 19.3 million and 10 million, respectively, in 2020. The global cancer burden is expected to be 28.4 million cases in 2040, a 47% rise from 2020 [1]. Well-recognized risk factors for cancer are tobacco use, poor physical activity, poor nutrition [2], obesity [3] and excessive alcohol consumption [4, 5]. The treatment of cancer includes three main modalities namely surgery, radiotherapy and chemotherapy. While the novel strategies may include targeted therapy, electric field treatments and vaccine therapy [6]. Chemotherapy is an intense and cyclic treatment with many side effects such as hair loss, nausea, vomiting and diarrhea [7]. Cyclic dependent kinase inhibitors that have emerged as a potent strategy for the treatment of advanced cancers [8]. The importance of chemotherapy for cure of cancer is increasing, especially with its use as adjuvants to local therapy. Besides, in advanced disease, chemotherapy has an expanding role in efforts to relieve cancer-related symptoms and to prolong life [9].

Cancer remains one of the most feared illnesses and the diagnosis of cancer has a huge psychological impact on the patients and their care-givers [10]. Many cancer patients suffer from psychological problems, such as depression. Cancer patients undergoing chemotherapy, usually experience various symptoms such as depression. Depression worsens during chemotherapy, persists for a long time after the end of chemotherapy, and it is also manifested in the recurrence of the disease [11]. This may interfere with the patient's ability to cope with the burden of the illness, it may decrease the acceptance of treatment, extend hospitalization [12], reduce the quality of life [13, 14], and increase suicide risk [15]. Furthermore, the depression compromises patient outcomes, resulting in higher rates of mortality [16–18], with estimates as high as a 26% greater mortality rate among patients with depressive symptoms and a 39% higher mortality rate among those with a diagnosis of major depression [19].

The epidemiology of depression among cancer patients was to 23.4% in China [20], 48.7% in Pakistan [21], 17% in Australia [22], 38.2% in Greece [23], 46.5% in Saudi Arabia [24], 67.7% in Rwandan [25], 25% in Addis Ababa-Ethiopia [26], and 58.4% in Gondar-Ethiopia [27]. Different studies showed several factors that are associated with depression such as cancer types [28], female sex [29], old age [14, 30], the duration of cancer [31] and type of treatment. Furthermore, a slightly higher incidence of depression was found among cancer patients who underwent chemotherapy than patients who had not received chemotherapy [32]. Besides, depressive symptoms were more prevalent across patients who were hospitalized compared with patients in the outpatient setting [33].

Regarding depression among cancer patients, most of the studies came from the developed world, and very limited studies addressed populations from low- and middle-income

countries. Therefore, this study aimed to assess the prevalence of depression among cancer patients at Tikur Anbessa Specialized Hospital in Ethiopia.

# Methods

## Study area, design and period

The study was conducted at Tikur Anbessa Specialized Hospital (TASH) which is located in Addis Ababa, which is the capital city of Ethiopia. TASH oncology unit was established in 2015 under the Federal Minister of Health. It is located at Addis Ababa, Lideta Sub–City. The oncology unit provides health services for all cancer patients attending TASH from Addis Ababa city, and different regional states of Ethiopia. The oncology unit at TASH is the largest referral site in the country, providing service to more than 60,000 patients annually. It is the sole oncology referral and the only radiotherapy center in Ethiopia. The cross-sectional study design was employed from 1$^{st}$ March to April 30, 2021.

**Source population.** All adult cancer patients who had followed up at TASH and treated with chemotherapy have been taken as the source population.

**Study population.** All adult cancer patients under chemotherapy treatment and follow up service during the data collection period.

## Eligibility criteria

**Inclusion criteria.** All adult cancer patients under chemotherapy and follow up at oncologic clinic in TASH during the data collection period.

**Exclusion criteria.** Patients who had communication or hearing impairments were excluded.

## Sample size calculation and sampling techniques

The required sample size was determined using a single population proportion formula having the following assumptions: value for the 95% CI (Zα/2 = 1.96), the proportion of depression (P = 50%), and margin of error (d = 5%) Then, by adding 10% of study subjects as non-response rate, the final sample size was 422. The study subjects were selected using the systematic random sampling technique after determining the sampling fraction (k = 900/420 = 2) and the first participant was selected using the lottery method.

## Study variables

**Dependent variable.** Depression (Yes/No).

**Independent variables.** Socio-demographic variables: Age, educational status, marital status, employment, and monthly income.

Psychosocial factors: Social and husband support, emotional violence, physical violence, and sexual violence.

Substance use: Use any substance like khat, alcohol, and cigarette.

Clinical characteristics: Stage of cancer, duration of the illness, comorbidity, type of cancer.

Family history of psychiatric illness: A family history (first-degree relatives) of psychiatric problems.

## Data collection tools and procedures

A structured interviewer-administered questionnaire was used to collect data from all participants. Socio-demographic, clinical, and psychosocial factors were assessed using predefined checklists. The social support level was assessed using the Oslo social support scale-3. The Oslo

social support-3 consists of three items assessing the level of social support. The sum score ranges from 3 to 14, with high values representing strong levels and low values representing poor levels of social support [34]. Depression was assessed using Patient Health Questionnaire-9 (PHQ-9). The PHQ-9 comprises nine items that can be scored from 0 (not at all) to 3 (nearly every day) and the total score ranges from 0 to 27 to measure depression severity [35, 36]. The standard PHQ9 cut off points 1–4, 5–9, 10–14, 15–19 and greater than or equal to 20 was considered as having minimal depression symptoms, minor depression, moderate depression, moderately severe and major depression (severe), respectively. Then, the total depression score was dichotomized, and those cancer patients who scored 10 and above were considered as having symptomatic depression. PHQ-9 is a validated tool in Ethiopia [37].

## Data analysis

The collected data were entered into Epi-data version 4.2 and analyzed by SPSS version 25. Bivariate logistic regression was done and, variables with p-value < 0.05 were entered in to a multivariate logistic model. In multivariate logistic regression analysis, adjusted odds ratio with a p-value of < 0.05 was considered statistically significant.

## Data quality control

The questionnaire was prepared in English and translated to Amharic and translated back to English before the data collection process. The data collection instrument was pre-tested on 5% of the sample to improve the language clarity and appropriateness of the data collection tools. The estimated time required, and necessary amendments were made after the piloting of the questionnaire. The data were collected by four BSc nursing professionals who were trained for one day on the techniques of data collection, purpose of the study, and ethical considerations. The researcher checked the accuracy, completeness and consistency of questionnaires completed by the data collectors to ensure the quality of data and visited the data collectors as many times as possible to check whether he/she collected the data appropriately. The Amharic version of PHQ-9 is validated in Ethiopia.

## Ethical approval and consent to participate

Ethical clearance was obtained from Addis Ababa University, College of Health Sciences, School of Nursing and Midwifery, Research and Ethics Review Committee (IRB protocol: AAU/CHS/NSG/0020). A collaboration letter for data collection was also obtained from TASH. Verbal informed consent was obtained from each participant. The objective of the study and methods of data collection were briefly clarified and explained for each participant, before enrolling any eligible study participants. Then, verbal informed consent was obtained from each participant. As the study does not impose any harm to the participants and from the experience, patients feel their confidentiality is secured more when they give verbal informed consent than written informed consent. To assure confidentiality, a code number was used instead of the participants' name or identification number.

## Operational definitions

Poor social support: cancer patients who scored 3–8 on the (Oslo-3) social support scale during cancer.

Moderate social support: cancer patients who scored 9–11 on the (Oslo-3) social support scale during cancer.

Strong social support: cancer patients who scored 12–14 on the (Oslo-3) social support scale during cancer.

## Result

### Socio-demographic characteristics of the participants

Four hundred twenty participants were included in the final analysis in this study with a non-response rate of 99.5%. Majority, 243 (57.9%) were female, 220 (52.4%) were in the age between 35 and 51 years and 103 (24.5%) were above the age of 52 years old. The mean and standard deviation age of the participants was 43.6 (SD±12.37) years, over 297 (69%) were married, and 119 (28.3%) completed secondary school education (Table 1).

### Clinical and behavioral characteristics of the study participants

The clinical characteristics of the study participants indicate that; 142 (33.8%) have breast cancer, 120 (28.6%) were on stage 2 cancer, 52 (12.4%) drunk alcohol but now quitted and 33 (7.9%) of participants quitted smoke. Of the total study participants, 6 (1.4%) study participants had a family history of known mental illness and 2(0.5%) had a diagnosis of chronic kidney disease (Table 2).

### Psychosocial factors (in the last 6 months) of cancer patients

From the total of participants, 53 (12.6%) responded that their family or close relatives had died and 30 (7.1%) participants responded as having died a spouse, parent, or child during the disease. 125 (29.8%) responded to having major financial crisis. 20(4.8%) responded to being sacked from their job (Table 3).

**Table 1. Socio-demographic characteristics of study participants among adult cancer patients at the oncology clinic of TASH, Addis Ababa, Ethiopia, 2021.**

| Variables | Category | Frequency | Percent |
|---|---|---|---|
| **Sex** | Female | 243 | 57.9 |
| | Male | 177 | 42.1 |
| **Age in years** | 18–34 | 97 | 23.1 |
| | 35–51 | 220 | 52.4 |
| | ≥52 | 103 | 24.5 |
| **Educational status** | Illiterate | 45 | 10.7 |
| | Read and write only | 77 | 18.3 |
| | Primary | 94 | 22.4 |
| | Secondary | 119 | 28.3 |
| | College/university | 85 | 20.2 |
| **Marital status** | Single | 47 | 11.2 |
| | Married | 290 | 69 |
| | Divorced | 16 | 3.8 |
| | Widowed/r | 67 | 16 |
| **Occupation** | Private work | 106 | 25.2 |
| | Civil servant | 90 | 21.4 |
| | House wife | 116 | 27.6 |
| | Others | 108 | 25.7 |
| **Income** | Low | 60 | 14.3 |
| | Medium | 113 | 26.9 |
| | High | 247 | 58.8 |

**Table 2. Clinical and behavioral characteristics of study participants among adult cancer patients at oncology clinic of TASH, Addis Ababa, Ethiopia, 2021.**

| Variables | Category | Frequency | Percent |
|---|---|---|---|
| **Stage of cancer** | Stage 1 | 28 | 6.7 |
| | Stage 2 | 99 | 23.6 |
| | Stage 3 | 120 | 28.6 |
| | Stage 4 | 173 | 41.6 |
| **Type of cancer** | Breast cancer | 156 | 37.1 |
| | Colon cancer | 53 | 12.6 |
| | Prostate cancer | 32 | 7.6 |
| | NPC | 35 | 8.3 |
| | Cervical cancer | 52 | 12.4 |
| | Lung cancer | 33 | 7.9 |
| | Bladder cancer | 12 | 2.9 |
| | Thyroid cancer | 6 | 1.4 |
| | Hematologic cancer | 12 | 2.9 |
| | Edwing cancer | 24 | 5.7 |
| | Other | 5 | 1.2 |
| **Months since diagnosis** | <6 | 203 | 48.3 |
| | 7–12 | 113 | 26.9 |
| | >12 | 104 | 24.8 |
| **Duration since start of chemotherapy** | 1–3 months | 190 | 45.2 |
| | 4–6 months | 110 | 26.2 |
| | >6 months | 120 | 28.6 |
| **Family history with known mental illness** | Yes | 6 | 1.4 |
| | No | 414 | 98.6 |
| **Hypertension** | Yes | 1 | 0.2 |
| | No | 419 | 99.8 |
| **Diabetes Mellitus** | Yes | 1 | 0.2 |
| | No | 419 | 99.8 |
| **Chronic kidney disease** | Yes | 2 | 0.5 |
| | No | 418 | 99.5 |
| **Cardiovascular disease** | Yes | 1 | 0.2 |
| | No | 419 | 99.8 |
| **Presence of comorbidities** | Yes | 17 | 4 |
| | No | 403 | 96 |
| **Chewing** | Currently chat chewing | 1 | 0.2 |
| | Previously drunker | 29 | 6.9 |
| | Never used | 390 | 92.9 |
| **Alcohol** | Currently drinker | 3 | 0.7 |
| | Previously drunker | 52 | 12.4 |
| | Never drank | 365 | 86.9 |
| **Smoking status** | Currently smoker | 1 | 0.2 |
| | Previously smoker | 33 | 7.9 |
| | Never smoke | 386 | 91.9 |

Note: presence of comorbidities = include hypertension, diabetes mellitus and kidney disease

Abbreviations: SD = standard deviation, chronic illness =, NPC = Naso pharengyal, other type of cancer include = Kaposi sarcoma, adenocarcinoma, and oropharia cancer

**Table 3. Psychosocial factors (in the last 6 months) of adult cancer patients receiving chemotherapy at TASH, Addis Ababa, Ethiopia, 2021.**

| Variable | Category | Frequency | Percent |
|---|---|---|---|
| Serious illness or injury during cancer | Yes | 7 | 1.7 |
| | No | 413 | 98.3 |
| Close relative serious illness or injury or assault | Yes | 9 | 2.1 |
| | No | 411 | 97.9 |
| Died spouse, parent or child | Yes | 30 | 7.1 |
| | No | 390 | 92.9 |
| Died family or close relatives | Yes | 53 | 12.6 |
| | No | 367 | 87.4 |
| | Yes | 125 | 29.8 |
| Major financial crisis | No | 295 | 70.2 |
| Sacked from job | Yes | 20 | 4.8 |
| | No | 400 | 95.2 |
| Unemployed/not able to work | Yes | 41 | 9.8 |
| | No | 379 | 90.2 |
| Separation due to marital difficulty | Yes | 11 | 2.6 |
| | No | 409 | 97.4 |
| Broken off a steady relationship | Yes | 11 | 2.6 |
| | No | 409 | 97.4 |
| Serious problems with close friend, neighbor /relative | Yes | 19 | 4.5 |
| | No | 401 | 95.5 |
| Lost/stolen property which mattered a lot | Yes | 20 | 4.8 |
| | No | 400 | 95.2 |
| Any problems with police/court | Yes | 11 | 2.6 |
| | No | 409 | 97.4 |
| Trauma by your wife or husband | Yes | 6 | 1.4 |
| | No | 414 | 98.6 |
| Forced sexual activity | Yes | 1 | 0.2 |
| | No | 419 | 99.8 |

## Social support among cancer patients

Of the total participants, 94 (22.4%), 207 (49.3%) and 119 (28.3%) had poor, moderate and strong social support, respectively (Fig 1).

## Prevalence of depression among cancer patients

The overall prevalence of depression among adult cancer patients was 33.1% (95% CI = 0.2858, 0.3761) (Fig 2).

Minimal symptoms of depression, minor depression, moderate depression, moderate severe depression, and severe depression were found to be 272 (64.8%), 9 (2.1%), 104 (24.8%), 28 (6.7%), and 7 (1.7%), respectively. A higher prevalence of depression was seen among patients with nasopharyngeal cancer (NPC) 51%, followed by thyroid cancer 50%, and breast cancer 42% (Fig 3).

## Factors associated to depression

In bivariate logistic regression analysis, education, income, type of cancer, the stage of cancer, serious illness or assault happened to close relatives, major financial problem, unemployed,

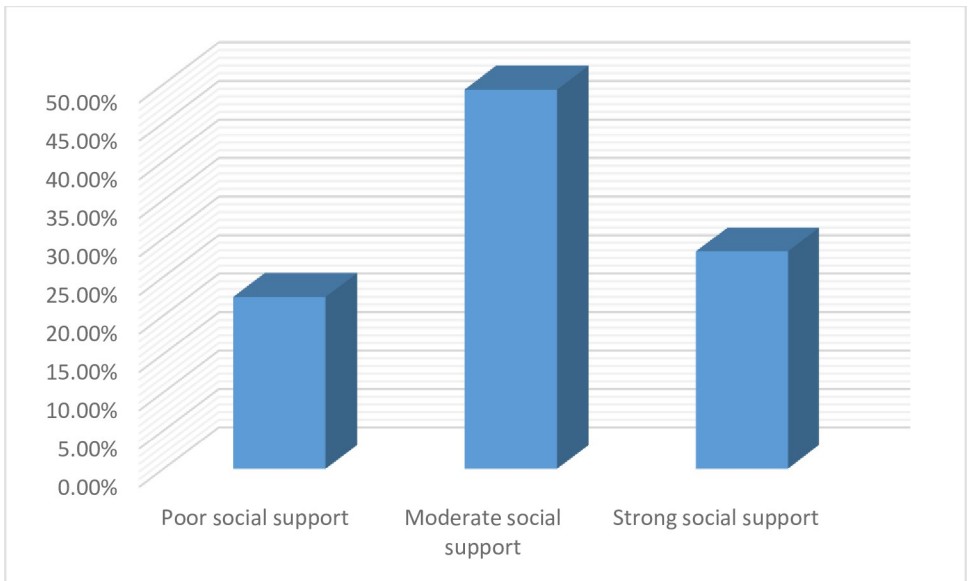

**Fig 1. Social support of adult cancer patients.**

sacked from a job, a serious problem with a close friend or neighborhood, having some problems with the police or courts, and social support were significantly associated with depression (p <0.05). When these variables were entered and analyzed in multivariate logistic regression, education, colon cancer, prostate cancer, cervical cancer, stage of cancer, unemployed and sacked from job were significantly associated with depression.

The results showed that participants who completed their primary education were less likely to develop depression than those who had no formal education (AOR: 0.414, 95% CI = 0.206–0.829). Additionally, patients with colon cancer were less likely to develop

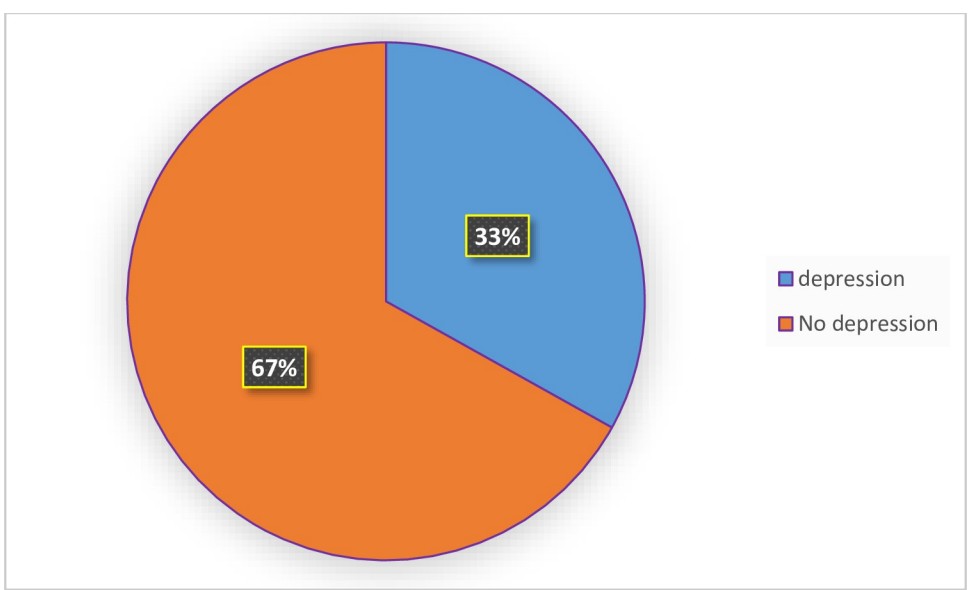

**Fig 2. Prevalence of depression among adult cancer patients.**

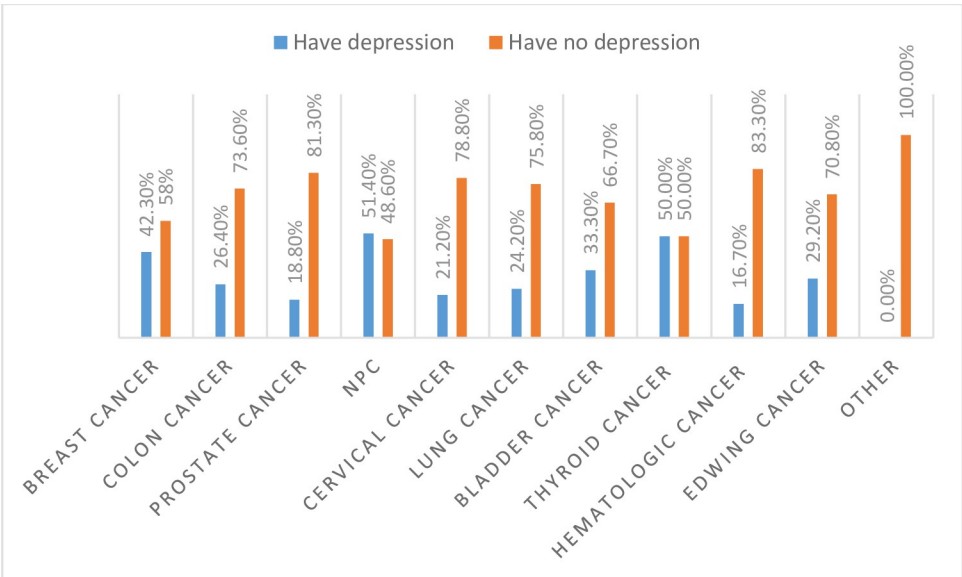

**Fig 3. Depression among each type of cancer.**

depression as compared to breast cancer (AOR: 0.364, 95% CI = 0.156–0.847). Besides, patients with prostate cancer and cervical cancer were less likely to develop depression compared to patients with breast cancer (AOR: 0.184, 95% CI = 0.064–0.525; AOR: 0.334, 95% CI = 0.148–0.752, respectively). Patients with Stage four cancer were seven times more likely to develop depression compared to patients with stage one (AOR: 7.444, 95% CI = 1.943–28.523). Being unemployed was two times more likely risk of developing depression compared to being employed (AOR: 2.263, 95% CI = 1.043–4.913), and patients sacked from job were three times more likely to develop depression compared to no sacke from job (AOR: 3.404, 95% CI = 1.049–11.042) (Table 4).

## Discussion

The overall prevalence of depression among adult cancer patients was 33.1% in this study. This finding was higher than the results of a studies conducted in Jordan-23.4% [38], India-22% [39], Iran-21.3% [31], Asia-29.6% [40], German-24% [41] and Ethiopia-Addis Ababa-25.0% [26]. However, it was lower than the findings of a study conducted in China- 66.7% [42], Ethiopia-Gondar-70.86% [43], Greece-38.2% [23], German 36.9% [44], Ethiopia 36.9% [45], and Rwanda 67.7% [46]. The variation might be due to differences in instruments to assess depression, criteria to define depression, and included cancer populations with respect to cancer type, stage and treatment modality.

Available data also suggest that cancer may increase patients' susceptibility to depression in several ways. To mention some, a reaction to a severe diagnosis and the forthcoming deterioration of health status may constitute a risk factor for depression; treatment with immune response modifiers and chemotherapy regimens, and experiencing of metabolic and endocrine alterations, chronic pain and extensive surgical interventions, may represent additional contributing factors [47, 48].

Patients who are at a high risk of depression require special attention and a strong support system. psychological problems hinder their ability to cope with treatment and symptoms, as well as recovery from chemotherapy side effects and cancer impact [49]. Depression in patients

**Table 4. Bivariate and multivariate logistic regression analysis of independent factors to depression among cancer patients at the oncology clinic of TASH, Addis Ababa, Ethiopia, 2021.**

| Variables | Category | Depression | | COR(95%CI) | AOR(95%CI) |
|---|---|---|---|---|---|
| | | No (n, %) | Yes (n, %) | | |
| **Education** | No formal education | 70(57.4) | 52(42.) | 1 | 1 |
| | Primary | 71(75.5) | 23(24.5) | 0.436(0.241–0.788)* | 0.414(0.206,0.829)* |
| | Secondary | 83(69.7) | 36(30.3) | 0.584(0.343–0.993)* | 0.538(0.279,1.037) |
| | Higher | 57(67.1) | 28(32.9) | 0.661(0.371–1.178) | 0.621(0.291,1.325) |
| **Income** | Low | 32(53.3) | 28(46.7) | 1.827(1.03,3.239)* | 1.211(0.573,2.56) |
| | Medium | 82(72.6) | 31(27.4) | 0.789(0.483–1.29) | 0.791(0.425,1.473) |
| | High | 167(67.6) | 80(32.4) | 1 | 1 |
| **Type of cancer** | Breast cancer | 90(57.7) | 66(42.3) | 1 | 1 |
| | Colon cancer | 39(73.6) | 14(26.4) | 0.49(0.246,0.974)* | 0.364(0.156,0.847)* |
| | Prostate cancer | 26(81.3) | 6(18.8) | 0.315(0.123,0.808)* | 0.184(0.064,0.525)* |
| | NPC | 17(48.6) | 18(51.4) | 1.444(0.692,3.011) | 1.408(0.609,3.256) |
| | Cervical cancer | 41(78.8) | 11(21.2) | 0.366(0.175,0.765)* | 0.334(0.148,0.752)* |
| | Lung cancer | 25(75.8) | 8(24.2) | 0.436(0.185,1.028) | 0.407(0.154,1.076) |
| | Bladder cancer | 8(66.7) | 4(33.3) | 0.682(0.197,2.36) | 0.775(0.196,3.065) |
| | Thyroid cancer | 3(50) | 3(50) | 1.364(0.267,6.97) | 1.291(0.19,8.791) |
| | Hematologic | 10(83.3) | 2(16.7) | 0.273(0.058,1.286) | 0.559(0.098,3.184) |
| | Edwing cancer | 17(70.8) | 7(29.2) | 0.561(0.22,1.431) | 0.617(0.215,1.766) |
| | Other | 5(100) | | - | - |
| **Stage of cancer** | Stage 1 | 25(89.3) | 3(10.7) | 1 | 1 |
| | Stage 2 | 81(81.8) | 18(19.2) | 1.852(0.504,6.808) | 1.893(0.461,7.764) |
| | Stage 3 | 85(70.8) | 35(29.2) | 3.431(0.973,12.104) | 3.558(0.904,14.009) |
| | Stage 4 | 90(52) | 83(48) | 7.685(2.237,26.402)* | 7.444(1.943,28.523)* |
| **Serious illness, injury or assault happened to a close relatives** | Yes | 35(51.5) | 33(48.5) | 2.188(1.291,3.707)* | 1.9(0.958,3.771) |
| | No | 246(69.9) | 106(30.1) | 1 | 1 |
| **A major financial crisis** | Yes | 80(56.7) | 61(43.3) | 0.509(0.333,0.777)* | 1.597(0.965,2.642) |
| | No | 201(72) | 78(28) | 1 | 1 |
| **Unemployed** | Yes | 16(39) | 25(61) | 3.632(1.868,7061)* | 2.263(1.043,4.913)* |
| | No | 165(69.9) | 114(30.1) | 1 | 1 |
| **Sacked from job** | Yes | 5(27.8) | 13(72.2) | 5.695(1.988,16.319)* | 3.404(1.049,11.042)* |
| | No | 276(68.7) | 126(31.3) | 1 | 1 |
| **A serious problem with a close friend, neighborhood** | Yes | 7(36.8) | 12(63.2) | 3.699(1.422, 9.617)* | 3.177(0.872,11.571) |
| | No | 274(68.3) | 127(31.7) | 1 | 1 |
| **Have you had any problems with the police or courts** | Yes | 4(36.4) | 7(63.6) | 3.672(1.057,12.765)* | 0.924(0.164,5.199) |
| | No | 277(67.7) | 132(32.3) | 1 | 1 |
| **Social support** | Poor | 91(56.9) | 69(43.1) | 1.896(1.106,3.248)* | 1.641(0.851,3.165) |
| | Moderate | 120(74.1) | 42(25.9) | 0.875(0.499,1.535) | 1.196(0.622,2.3) |
| | Strong | 70(71.4) | 28(28.6) | 1 | 1 |

NPC: Naso Pharyngeal Cancer; N: number; COR = Crud Odds ratio; AOR = Adjusted Odds Ratio; CI = Confidence Interval; %: percent

* = statistically significant by univariate logistic regression

** = statistically significant by multivariate logistic regression

with cancer can be prevented and prophylactic treatment should be administered during oncological treatment [50]. The treatment of depression must be started at an early stage. It can take a long time to make patients aware of the psychological nature of their difficulties,

having this consideration, encourage them to request a psychological consultation or to accept the prescription of psychotropic or, in particular, antidepressant medicaments [51].

The current study found several associated factors for depression among cancer patients in Ethiopia. Those associated factors were primary education, unemployment, sacked from jobs, colon cancer, prostate cancer, cervical cancer and cancer stage. Study participants who completed their primary education were less likely to develop depression than who had no formal education. This finding was in a line with the findings of studies conducted in Ethiopia [21, 26, 43]. Education may offer opportunities for developing interventions to reduce the disease burden of depression. Low education levels are linked with depression and that higher levels of education play a preventive role against depression [21]. Being unemployed was found to have 2.2 times a more likely risk to develop depression compared to being employed, which is supported by other studies [43, 52]. Patients sacked from job were 3.4 times more likely to develop depression compared to no sack from job. Having no or low education level, unemployed status, and being sacked from jobs, negatively influence depression symptoms. These often make the adaption process of psychological disorders to be longer and harder [53]. The experience of being diagnosed with cancer could become a source of distress in addition to the disruption in the work role. Failure to remain in employment could deprive an individual from social contact and well-being.

Patients who had colon, prostate and cervical cancer were less likely to have depression compared with those with breast cancer. This finding was similar in other studies [54]. The possible explanation for this occurrence might be increased worriedness of patients after breast cancer diagnosis, to shortened survival time, recurrence and metastasis. Furthermore, female patients face impaired body image and decreased quality of sexual life caused by surgery and other treatments. Therefore, compared with other malignant tumors, the incidence of depression and other psychiatric symptoms in breast cancer patients are higher, which is also closely related to the lack of female secondary characteristics and physical symptoms such as nausea, vomiting, fatigue, hair loss, and insomnia caused by chemotherapy [55].

Moreover, patients with stage four had 7.4 times more likely at risk of developing depression compared to patients with stage one. Similar findings have been found in other studies [14, 46]. This might be due to when the stage is advanced, patients may fear or stress. Studies found that individuals diagnosed at an advanced stage have a higher risk of developing depression compared with those diagnosed at an early stage [14, 56]. Patients with advanced-stage cancer generally have a greater need for supportive care, due to poor physical functioning, a greater symptom burden, and higher levels of distress, and depression than other cancer patients [57]. Earlier and more intensive supportive care for patients with cancer reduces symptom burden and may prolong life for patients with advanced disease [58]. For managing their distress, pharmacotherapy and psychotherapy are effective for managing depression among advanced cancer patients [59].

The preventive strategies toward the risk factors of depression include education and counselling programs that must be designed and planned according to the patient's wishes and learning needs [60]. The current study is conducted in light with some limitations. First, some of the depression symptoms may have been due to the cancer itself or its treatment rather than due to depression.

## Conclusion

Depression is found to be a major public health problem for adult cancer patients in Ethiopia. Those cancer patients who had unemployed status, sacked from jobs, and stage IV cancer were most likely to develop depression, whereas patients having completed primary education,

colon, prostate and cervical cancer were less likely to have depression. To reduce the prevalence of depression among cancer patients, designing strategies and special attention are needed to bring changes in the psychological status of patients in advanced disease stage.

## Supporting information

**S1 Text. Sociodemographic questions.**
(DOCX)

**S2 Text. Clinical and behavioral characteristics of questions.**
(DOCX)

**S1 Table. Psychosocial questions.**
(DOCX)

**S2 Table. Socila support questions.**
(DOCX)

**S3 Table. Tools used to assess depression.**
(DOCX)

**S1 File. Data collection tool.**
(DOCX)

## Acknowledgments

First of all, we want to give appreciation to Addis Ababa University, College of Health Science post graduate office for giving us this chance to participate in the research activity. We would like to thank the patients, data collectors, and supervisors who were involved in this study and spent their valuable time responding to our study.

## Author Contributions

**Conceptualization:** Abebe Muche Belete, Anmut Alemagegn, Anemut Tilahun Mulu, Taklo Simeneh Yazie, Bekalu Bewket, Adisu Asefa, Wendimeneh Shibabaw Shiferaw.

**Data curation:** Bekalu Bewket.

**Formal analysis:** Abebe Muche Belete, Bekalu Bewket, Adisu Asefa, Wendimeneh Shibabaw Shiferaw.

**Methodology:** Abebe Muche Belete, Anmut Alemagegn, Anemut Tilahun Mulu, Taklo Simeneh Yazie, Bekalu Bewket, Adisu Asefa, Wendimeneh Shibabaw Shiferaw.

**Validation:** Abebe Muche Belete, Anmut Alemagegn.

**Visualization:** Anemut Tilahun Mulu, Taklo Simeneh Yazie, Bekalu Bewket, Adisu Asefa, Wendimeneh Shibabaw Shiferaw.

**Writing – original draft:** Abebe Muche Belete, Taklo Simeneh Yazie, Bekalu Bewket.

**Writing – review & editing:** Anemut Tilahun Mulu.

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
