## [Decision Letter · Decision Letter 0]

4 Apr 2022

PONE-D-22-06018Prevalence of depression and associated factors among adult cancer patients receiving chemotherapy during the era of COVID-19 in Ethiopia. A Hospital based cross sectional studyPLOS ONE

Dear Dr. Belete,

Thank you for submitting your manuscript to PLOS ONE. After careful consideration, we feel that it has merit but does not fully meet PLOS ONE’s publication criteria as it currently stands. Therefore, we invite you to submit a revised version of the manuscript that addresses the points raised during the review process.

I have received the reports from our advisors on your manuscript which you submitted to PLOS ONE.

Based on the comments received, I feel that your manuscript could be reconsidered for publication should you be prepared to incorporate major revisions.

When preparing your revised manuscript, you are asked to carefully consider the reviewer comments below and submit a list of responses to the comments.

Editor Comments: The paper should be checked by a professional speaker of English before complete acceptance.

We look forward to receiving your revised manuscript.

Kind regards,

Muhammad Sajid Hamid Akash

Academic Editor

PLOS ONE

Journal Requirements:

When submitting your revision, we need you to address these additional requirements. 1. Please ensure that your manuscript meets PLOS ONE's style requirements, including those for file naming. The PLOS ONE style templates can be found at https://journals.plos.org/plosone/s/file?id=wjVg/PLOSOne_formatting_sample_main_body.pdf and https://journals.plos.org/plosone/s/file?id=ba62/PLOSOne_formatting_sample_title_authors_affiliations.pdf
 2. We suggest you thoroughly copyedit your manuscript for language usage, spelling, and grammar. If you do not know anyone who can help you do this, you may wish to consider employing a professional scientific editing service.  Whilst you may use any professional scientific editing service of your choice, PLOS has partnered with both American Journal Experts (AJE) and Editage to provide discounted services to PLOS authors. Both organizations have experience helping authors meet PLOS guidelines and can provide language editing, translation, manuscript formatting, and figure formatting to ensure your manuscript meets our submission guidelines. To take advantage of our partnership with AJE, visit the AJE website (http://learn.aje.com/plos/) for a 15% discount off AJE services. To take advantage of our partnership with Editage, visit the Editage website (www.editage.com) and enter referral code PLOSEDIT for a 15% discount off Editage services.  If the PLOS editorial team finds any language issues in text that either AJE or Editage has edited, the service provider will re-edit the text for free. Upon resubmission, please provide the following: The name of the colleague or the details of the professional service that edited your manuscript A copy of your manuscript showing your changes by either highlighting them or using track changes (uploaded as a *supporting information* file) A clean copy of the edited manuscript (uploaded as the new *manuscript* file)” 3. Please provide additional details regarding participant consent. In the ethics statement in the Methods and online submission information, please ensure that you have specified (1) whether consent was informed and (2) what type you obtained (for instance, written or verbal, and if verbal, how it was documented and witnessed). If your study included minors, state whether you obtained consent from parents or guardians. If the need for consent was waived by the ethics committee, please include this information. If you are reporting a retrospective study of medical records or archived samples, please ensure that you have discussed whether all data were fully anonymized before you accessed them and/or whether the IRB or ethics committee waived the requirement for informed consent. If patients provided informed written consent to have data from their medical records used in research, please include this information. 4. Thank you for stating the following financial disclosure:  "The funders had no role in study design, data collection and analysis, decision to publish, or preparation of the manuscript." At this time, please address the following queries: a) Please clarify the sources of funding (financial or material support) for your study. List the grants or organizations that supported your study, including funding received from your institution. b) State what role the funders took in the study. If the funders had no role in your study, please state: “The funders had no role in study design, data collection and analysis, decision to publish, or preparation of the manuscript.”c) If any authors received a salary from any of your funders, please state which authors and which funders.d) If you did not receive any funding for this study, please state: “The authors received no specific funding for this work.” Please include your amended statements within your cover letter; we will change the online submission form on your behalf. 5. PLOS requires an ORCID iD for the corresponding author in Editorial Manager on papers submitted after December 6th, 2016. Please ensure that you have an ORCID iD and that it is validated in Editorial Manager. To do this, go to ‘Update my Information’ (in the upper left-hand corner of the main menu), and click on the Fetch/Validate link next to the ORCID field. This will take you to the ORCID site and allow you to create a new iD or authenticate a pre-existing iD in Editorial Manager. Please see the following video for instructions on linking an ORCID iD to your Editorial Manager account: https://www.youtube.com/watch?v=_xcclfuvtxQ

Reviewers' comments:

Reviewer's Responses to Questions

**Comments to the Author**

1. Is the manuscript technically sound, and do the data support the conclusions?

Reviewer #1: Partly

2. Has the statistical analysis been performed appropriately and rigorously? 

Reviewer #1: Yes

3. Have the authors made all data underlying the findings in their manuscript fully available?

Reviewer #1: Yes

4. Is the manuscript presented in an intelligible fashion and written in standard English?

Reviewer #1: Yes

5. Review Comments to the Author

Reviewer #1: The author presented an original article on ‘Prevalence of depression and associated factors among adult cancer patients receiving chemotherapy during the era of COVID-19 in Ethiopia. A Hospital based cross sectional study’. The manuscript needs to be improved regarding the following aspects.

• Cancer chemotherapy should be described in detail to strengthen your research study moreover, overall associated factors for cancer risk should also be included in the introduction part, Biomed Pharmacother. 2021; 143: 112119. https://doi.org/10.1016/j.biopha.2021.112119

• In the discussion section, discuss more details of the results and their outcome risks and preventive strategies.

• More description is required to explain the association between key points and the conclusion.

• In this article not a single graphical representation has been included, add approximately 3 to 4 figures or graphs to strengthen your research manuscript.

• There are a few formatting mistakes, for this deeply review the article.

• Many references are missing or inconsistent in format.

• There are some grammatical mistakes in the manuscript, such as verbs and prepositions. The manuscript needs extensive review by an author.

6. PLOS authors have the option to publish the peer review history of their article (what does this mean?). If published, this will include your full peer review and any attached files.

Reviewer #1: No

---

## [Author Response · Author response to Decision Letter 0]

24 Apr 2022

PONE-D-22-06018

Prevalence of depression and associated factors among adult cancer patients receiving chemotherapy during the era of COVID-19 in Ethiopia. A Hospital based cross sectional study

PLOSONE

Dear Dr. Belete,

Thank you for submitting your manuscript to PLOS ONE. After careful consideration, we feel that it has merit but does not fully meet PLOS ONE’s publication criteria as it currently stands. Therefore, we invite you to submit a revised version of the manuscript that addresses the points raised during the review process.

Response: Dear editor, thank you very much for allowing us to revise our manuscript. We have tried to respond to each concern as much as possible. We have submitted a revised version of the manuscript.

I have received the reports from our advisors on your manuscript which you submitted to PLOS ONE.

Based on the comments received, I feel that your manuscript could be reconsidered for publication should you be prepared to incorporate major revisions. When preparing your revised manuscript, you are asked to carefully consider the reviewer comments below and submit a list of responses to the comments.

Author response: Dear editor, thank you for considering our manuscript publication. We have addressed the points raised by the reviewer and editors. 

Editor Comments: The paper should be checked by a professional speaker of English before complete acceptance. 

Author response: Dear editor, thank you very much. We have tried to edit the manuscript and also we used the online Grammarlly using the following links: https://www.grammarly.com/

Author response: Dear editor, thank you. We have included ‘Response to Reviewers', 'Revised Manuscript with Track Changes’, and ‘Manuscript’ separately. 

Author response: Dear editor, thanks for allowing us to update the financial disclosure. We have added it to the cover letter. Thanks for the suggestion for guidelines for figures.

If applicable, we recommend that you deposit your laboratory protocols in protocols.io to enhance the reproducibility of your results. Protocols.io assigns your protocol its own identifier (DOI) so that it can be cited independently in the future. For instructions see: https://journals.plos.org/plosone/s/submission-guidelines#loc-laboratory-protocols. Additionally, PLOS ONE offers an option for publishing peer-reviewed Lab Protocol articles, which describe protocols hosted on protocols.io. Read more information on sharing protocols at https://plos.org/protocols?utm_medium=editorial email&utm_source=authorletters&utm_campaign=protocols.

Author response: Dear editor, thank you for recommending depositing the laboratory protocols in protocols.io. But it is not applicable

We look forward to receiving your revised manuscript. 

Kind regards, 

Muhammad Sajid Hamid Akash 

Academic Editor PLOS ONE

Journal Requirements:

Author response: Dear Editor, we saw the link you invited me and I made corrections to the manuscript style to meet the requirement of PLOS ONE. 

Author response: Dear editor, thank you for your suggestion. 

Author response: Dear Editor, we edit the manuscript by using an online grammar checker.

Author response: Dear Editor, we have included manuscripts that contain track changes and uploaded them as manuscript track changes.

Author response: Dear Editor, thank you. We have included a clean copy of the edited manuscript and uploaded them as a manuscript file.

Author response: Dear Editor, 1) consent was informed 2) the consent obtained was verbal informed consent. Detail of the study’s objective, method of data collection, and ethical concern were explained to potential participants, and they were requested to participate in the study voluntarily. For volunteer participants, data collectors tick the right sign in front of I agree, then continue interviewing the participant. There were no minors in this study, all were adults who can themselves give consent after understanding what was explained about the study. The study participants gave verbal consent to use their medical cards. 

Author response: Dear Editor, sorry for making confused. It is a technical error this work had not received any form of support. All cost is covered by author. 

State what role the funders took in the study. If the funders had no role in your study, please state: “The funders had no role in study design, data collection and analysis, decision to publish, or preparation of the manuscript.”

Author response: Dear Editor, again sorry. This study did not receive any form of support. Therefore, there is no funder.

b) If any authors received a salary from any of your funders, please state which authors and which funders.

Author response: Dear Editor, since no funder was participated so no author is received salary from funders.

 Author response: Dear Editor, the authors received no specific funding for this work.

Author response: Dear Editor, thank you very much. The amended statement is within the cover letter.

Author response: Dear Editor, thank you for your recommendation. Already I have an ORCID id

Reviewers' comments:

Reviewer's Responses to Questions

Comments to the Author

1. Is the manuscript technically sound, and do the data support the conclusions?

Reviewer #1: Partly

Author response: Dear reviewer, we have tried to modify it. See the revised track changes. 

2. Has the statistical analysis been performed appropriately and rigorously? 

Reviewer #1: Yes

Author response: Dear reviewer, thank you very much.

3. Have the authors made all data underlying the findings in their manuscript fully available?

Reviewer #1: Yes

Author response: Dear reviewer, thank you

4. Is the manuscript presented in an intelligible fashion and written in standard English?

Reviewer #1: Yes

Author response: Dear reviewer, thank you very much

5. Review Comments to the Author

Response: Dear editor, thank you very much. We have tried to answer the reviewer’s feedback and question as much as we can.

Reviewer #1: The author presented an original article on ‘Prevalence of depression and associated factors among adult cancer patients receiving chemotherapy during the era of COVID-19 in Ethiopia. A Hospital based cross sectional study’. The manuscript needs to be improved regarding the following aspects.

• Cancer chemotherapy should be described in detail to strengthen your research study moreover, overall associated factors for cancer risk should also be included in the introduction part, Biomed Pharmacother. 2021; 143: 112119. https://doi.org/10.1016/j.biopha.2021.112119

Author response: Dear reviewer, thank you for your recommendation. As per your recommendation, we have made an amendment to the revised manuscript. 

• In the discussion section, discuss more details of the results and their outcome risks and preventive strategies.

Author response: Dear reviewer, thank you very much. We have tried to address all the points you raised in the revised manuscript.

• More description is required to explain the association between key points and the conclusion.

Author response: Dear reviewer, we have tried to add more description in the discussion part that links to a conclusion. 

• In this article not a single graphical representation has been included, add approximately 3 to 4 figures or graphs to strengthen your research manuscript.

Author response: Dear reviewer, thank you very much for your effort to see each part of our work. According to your suggestion, we add 3 figures to the revised manuscript. 

• There are a few formatting mistakes, for this deeply review the article.

Author response: Dear reviewer, we review deeply and modification was done in the revised manuscript. 

• Many references are missing or inconsistent in format.

Author response: Dear reviewer, I made corrections as per your comment. 

• There are some grammatical mistakes in the manuscript, such as verbs and prepositions. The manuscript needs extensive review by an author.

Author response: Dear reviewer, as per your comment I revise the manuscript for grammatical and spelling errors. The corrections are highlighted in the document entitled “revised manuscript with track changes”.

6. PLOS authors have the option to publish the peer review history of their article (what does this mean?). If published, this will include your full peer review and any attached files.

Do you want your identity to be public for this peer review? For information about this choice, including consent withdrawal, please see our Privacy Policy.

Reviewer #1: No

Author response: Dear editor, thank you for your suggestion. We used the link that you invite me.

---

## [Decision Letter · Decision Letter 1]

2 Jun 2022

PONE-D-22-06018R1Prevalence of depression and associated factors among adult cancer patients receiving chemotherapy during the era of COVID-19 in Ethiopia. Hospital-based cross-sectional studyPLOS ONE

Dear Dr. Belete,

Thank you for submitting your manuscript to PLOS ONE. After careful consideration, we feel that it has merit but does not fully meet PLOS ONE’s publication criteria as it currently stands. Therefore, we invite you to submit a revised version of the manuscript that addresses the points raised during the review process.

I have received the reports from our advisors on your manuscript which you submitted to PLOS ONE.

Based on the comments received, I feel that your manuscript could be reconsidered for publication should you be prepared to incorporate major revisions.

When preparing your revised manuscript, you are asked to carefully consider the reviewer comments below and submit a list of responses to the comments.

Editor Comments: There is a huge list of grammatical mistakes and syntax errors. The paper should be checked by a professional speaker of English before complete acceptance.

We look forward to receiving your revised manuscript.

Kind regards,

Muhammad Sajid Hamid Akash

Academic Editor

PLOS ONE

Reviewers' comments:

Reviewer's Responses to Questions

**Comments to the Author**

1. If the authors have adequately addressed your comments raised in a previous round of review and you feel that this manuscript is now acceptable for publication, you may indicate that here to bypass the “Comments to the Author” section, enter your conflict of interest statement in the “Confidential to Editor” section, and submit your "Accept" recommendation.

Reviewer #1: (No Response)

2. Is the manuscript technically sound, and do the data support the conclusions?

Reviewer #1: Partly

3. Has the statistical analysis been performed appropriately and rigorously? 

Reviewer #1: Yes

4. Have the authors made all data underlying the findings in their manuscript fully available?

Reviewer #1: Yes

5. Is the manuscript presented in an intelligible fashion and written in standard English?

Reviewer #1: Yes

6. Review Comments to the Author

Reviewer #1: The author presented an original article on “Prevalence of depression and associated factors among adult cancer patients receiving chemotherapy during the era of COVID-19 in Ethiopia. Hospital-based cross-sectional study”. The manuscript needs to be improved regarding the following aspects.

• The literature part is not up-to-date as I recommended you make changes in your manuscript that will strengthen your work. Again, I asked you to update the literature section by briefly explaining the cancer-associated risk factors and some of its novel therapeutic interventions. Biomed Pharmacother. 2021; 143: 112119. https://doi.org/10.1016/j.biopha.2021.112119. Biomed Pharmacother. 2018; 107: 1326-41. https://doi.org/10.1016/j.biopha.2018.08.116.

• The figures you have added are quite simple, more work is needed on graphics. Kindly revise them (figure1 and figure 2) with more creativity and diversity.

7. PLOS authors have the option to publish the peer review history of their article (what does this mean?). If published, this will include your full peer review and any attached files.

Reviewer #1: No

---

## [Author Response · Author response to Decision Letter 1]

5 Jun 2022

PONE-D-22-06018R1

Prevalence of depression and associated factors among adult cancer patients receiving chemotherapy during the era of COVID-19 in Ethiopia. Hospital-based cross-sectional study

PLOS ONE

Dear Dr. Belete

Thank you for submitting your manuscript to PLOS ONE. After careful consideration, we feel that it has merit but does not fully meet PLOS ONE’s publication criteria as it currently stands. Therefore, we invite you to submit a revised version of the manuscript that addresses the points raised during the review process.

Response: Thank you very much for allowing us to revise our manuscript again. We have tried to response each concerns as much as possible. We have submitted a revised version of the manuscript.

I have received the reports from our advisors on your manuscript which you submitted to PLOS ONE.

Based on the comments received, I feel that your manuscript could be reconsidered for publication should you be prepared to incorporate major revisions.

When preparing your revised manuscript, you are asked to carefully consider the reviewer comments below and submit a list of responses to the comments.

Response: Again thank you very much. We will provide a list of response. 

Editor Comments: There is a huge list of grammatical mistakes and syntax errors. The paper should be checked by a professional speaker of English before complete acceptance.

Response: we have extensively edited the manuscript for language usage, spelling, and grammar before submitting the revised version. The formatting of the text and document (text sizes and grammatical errors) were also edited. We have edited using the free editing chance at Trinka. Using the following link: https://www.trinka.ai

Additionally, we have extensively edited the manuscript by a professional language editor, (English-language speaker thoroughly edited the manuscript for language usage, spelling, and grammar) before submitting the revised version. His name is Dr. Menakath Menon.

Response: Thank you. We have included ‘Response to Reviewers', 'Revised Manuscript with Track Changes and Manuscript separately 

Response: thank you for reminding us on financial disclosure. The updated financial disclosure is given in the cover letter. 

If applicable, we recommend that you deposit your laboratory protocols in protocols.io to enhance the reproducibility of your results. Protocols.io assigns your protocol its own identifier (DOI) so that it can be cited independently in the future. For instructions see: https://journals.plos.org/plosone/s/submission-guidelines#loc-laboratory-protocols. Additionally, PLOS ONE offers an option for publishing peer-reviewed Lab Protocol articles, which describe protocols hosted on protocols.io. Read more information on sharing protocols at https://plos.org/protocols?utm_medium=editorial-email&utm_source=authorletters&utm_campaign=protocols

Response: thank you for your suggestion, but we haven’t any form of laboratory protocols. 

Reviewer's Responses to Questions

Comments to the Author

1. If the authors have adequately addressed your comments raised in a previous round of review and you feel that this manuscript is now acceptable for publication, you may indicate that here to bypass the “Comments to the Author” section, enter your conflict of interest statement in the “Confidential to Editor” section, and submit your "Accept" recommendation.

Reviewer #1: (No Response)

Response: sorry for inadequately addressed your concern. However, in the current version, all of your concern is updated as per your comments.

2. Is the manuscript technically sound, and do the data support the conclusions?

Reviewer #1: Partly

Response: in fact your concern is appreciated. We have made extensive modification in the revised manuscript.

3. Has the statistical analysis been performed appropriately and rigorously?

Reviewer #1: Yes

Response: Thank you.

4. Have the authors made all data underlying the findings in their manuscript fully available?

Reviewer #1: Yes

Response: Thank you

5. Is the manuscript presented in an intelligible fashion and written in standard English?

Reviewer #1: Yes

Response: Thank you.

6. Review Comments to the Author

Response: thanks .We tried to address each and every question/concern step by step

Reviewer #1: The author presented an original article on “Prevalence of depression and associated factors among adult cancer patients receiving chemotherapy during the era of COVID-19 in Ethiopia. Hospital-based cross-sectional study”. The manuscript needs to be improved regarding the following aspects.

• The literature part is not up-to-date as I recommended you make changes in your manuscript that will strengthen your work. Again, I asked you to update the literature section by briefly explaining the cancer-associated risk factors and some of its novel therapeutic interventions. Biomed Pharmacother. 2021; 143: 112119. https://doi.org/10.1016/j.biopha.2021.112119. Biomed Pharmacother. 2018; 107: 1326-41. https://doi.org/10.1016/j.biopha.2018.08.116.

Response: Thank you for providing the most updated article. As per your suggestion we made modification in the revised manuscript. 

• The figures you have added are quite simple, more work is needed on graphics. Kindly revise them (figure1 and figure 2) with more creativity and diversity.

Response: we tried to make more attractive.

7. PLOS authors have the option to publish the peer review history of their article (what does this mean?). If published, this will include your full peer review and any attached files.

Do you want your identity to be public for this peer review? For information about this choice, including consent withdrawal, please see our Privacy Policy.

Reviewer #1: No

Response: thank you

---

## [Editor Report · Decision Letter 2]

8 Jun 2022

Prevalence of depression and associated factors among adult cancer patients receiving chemotherapy during the era of COVID-19 in Ethiopia. Hospital-based cross-sectional study

PONE-D-22-06018R2

Dear Dr. Belete,

We’re pleased to inform you that your manuscript has been judged scientifically suitable for publication and will be formally accepted for publication once it meets all outstanding technical requirements.

Kind regards,

Muhammad Sajid Hamid Akash

Academic Editor

PLOS ONE
---

## [Editor Report · Acceptance letter]

17 Jun 2022

PONE-D-22-06018R2 

Prevalence of depression and associated factors among adult cancer patients receiving chemotherapy during the era of COVID-19 in Ethiopia. Hospital-based cross-sectional study 

Dear Dr. Belete:

I'm pleased to inform you that your manuscript has been deemed suitable for publication in PLOS ONE. Congratulations! Your manuscript is now with our production department. 

Kind regards, 

on behalf of

Dr. Muhammad Sajid Hamid Akash 

Academic Editor

PLOS ONE